# The triad interaction of ULK1, ATG13, and FIP200 is required for ULK complex formation and autophagy

Yutaro Hama[1,2,3], Yuko Fujioka[1], Hayashi Yamamoto[3,4], Noboru Mizushima[3], Nobuo N Noda[1,2]*

[1]Institute for Genetic Medicine, Hokkaido University, Sapporo, Japan; [2]Institute of Microbial Chemistry, Tokyo, Japan; [3]Department of Biochemistry and Molecular Biology, Graduate School of Medicine, The University of Tokyo, Tokyo, Japan; [4]Department of Molecular Oncology, Nippon Medical School, Institute for Advanced Medical Sciences, Tokyo, Japan

## eLife Assessment

Building on previous structural studies, this work provides **valuable** new insights into the architecture of the autophagy initiation complex, comprising ULK1, ATG13, and FIP200. The authors present their findings with **solid** supporting evidence, making this study a significant contribution to the autophagy field.

*For correspondence: nn@igm.hokudai.ac.jp

## Abstract

In mammals, autophagosome formation, a central event in autophagy, is initiated by the ULK complex comprising ULK1/2, FIP200, ATG13, and ATG101. However, the structural basis and mechanism underlying the ULK complex assembly have yet to be fully clarified. Here, we predicted the core interactions organizing the ULK complex using AlphaFold, which proposed that the intrinsically disordered region of ATG13 engages the bases of the two UBL domains in the FIP200 dimer via two phenylalanines and also binds the tandem microtubule-interacting and transport domain of ULK1, thereby yielding the 1:1:2 stoichiometry of the ULK1–ATG13–FIP200 complex. We validated the predicted interactions by point mutations and demonstrated direct triad interactions among ULK1, ATG13, and FIP200 in vitro and in cells, wherein each interaction was additively important for autophagic flux. These results indicate that the ULK1–ATG13–FIP200 triadic interaction is crucial for autophagosome formation and provides a structural basis and insights into the regulation mechanism of autophagy initiation in mammals.

## Introduction

Macroautophagy (hereafter autophagy) is an intracellular degradation system delivering cytoplasmic components to the lysosome via autophagosomes (*Mizushima and Komatsu, 2011*). Autophagy contributes to intracellular quality control by degrading defective proteins and organelles, such as polyubiquitinated proteins and damaged mitochondria. Alternatively, autophagy is induced by starvation as an adaptation mechanism. These physiological roles of autophagy are associated with various diseases, including cancer and neurodegenerative disorders (*Mizushima and Levine, 2020*). Therefore, understanding and controlling autophagy in clinical contexts is essential.

Autophagosome formation is performed by the evolutionarily conserved autophagy-related (ATG) proteins, which form several functional units to act directly on autophagosome formation (*Nakatogawa, 2020*). Among them, the Atg1/ULK complex is critical in initiating autophagosome formation

by organizing the formation site where all the functional units, such as ATG9 vesicles and ATG8 conjugation system, colocalize (*Hama et al., 2023*; *Kannangara et al., 2021*; *Kishi-Itakura et al., 2014*; *Ren et al., 2023*; *Suzuki et al., 2001*; *Yamamoto et al., 2012*; *Zhou et al., 2021*). In mammals, the ULK complex includes ULK1/2, ATG13, ATG101, and RB1CC1/FIP200 (*Mizushima, 2010*; *Wong et al., 2013*). Structural and functional analysis has made considerable progress in the yeast counterpart, the Atg1 complex, whose core comprises Atg1 (ULK1/2 homolog), Atg13, and Atg17 (remote homolog of FIP200) (*Fujioka et al., 2014*; *Li et al., 2014 Lin and Hurley, 2016*; *Noda, 2024*; *Ragusa et al., 2012*). Crystallographic analyses have unveiled the detailed interaction mode of Atg13 with Atg1 and Atg17 and the mechanism of the Atg1 complex organization (*Fujioka et al., 2014*). Moreover, structure-based in vitro and in vivo functional analyses have revealed that the Atg1 complex undergoes liquid–liquid phase separation to organize the autophagosome formation site on the vacuolar membrane to initiate autophagy (*Fujioka et al., 2020*). However, structural and functional analyses of the mammalian ULK complex lag behind that of the yeast Atg1 complex, and structural information on the interactions comprising the core of the ULK complex has long been limited to low-resolution electron microscopy (EM) data (*Shi et al., 2020*). A recent preprint reported the cryo-EM structures of the human ULK1 core complex and the supercomplex between the ULK1 core complex and the class III phosphatidylinositol 3-kinase complex 1 at 4.2–6.84 Å resolution, which discovered the unprecedented interaction between these two critical complexes working at autophagy initiation (*Chen et al., 2023*). Moreover, the local resolution of 3.35 Å enabled them to resolve the ULK1-mediated interactions in the ULK1 core complex. However, not all of the interactions that construct the ULK1 core complex have been perfectly resolved. Recently, calcium transients on the endoplasmic reticulum surface were proposed to trigger the liquid–liquid phase separation of FIP200 to organize the autophagosome formation site, thereby initiating autophagy (*Zheng et al., 2022*). However, the lack of structural information makes elucidating the molecular mechanisms underlying these events challenging.

This study predicted the structure of the ULK1–ATG13–FIP200 complex using the AlphaFold2 multimer (*Evans et al., 2021*), which unveiled the detailed interactions between ATG13 and FIP200, ATG13 and ULK1, and ULK1 and FIP200. In vitro and in vivo mutational analysis confirmed all the predicted interactions, which were demonstrated to be important for autophagy. Furthermore, we found that the FIP200–ATG13 and ULK1–ATG13 interactions partially complement each other for autophagy initiation. These findings establish the mechanism of ULK complex formation and provide a structural basis for understanding the ULK complex phase separation and regulation mechanism of autophagy in the mammalian context.

## Results

### Structural prediction of the ULK1–ATG13–FIP200 complex identifies FIP200-interacting residues in ATG13

Previous studies reported that the C-terminal region of ULK1 and the C-terminal intrinsically disordered region of ATG13 bind to the N-terminal region of the FIP200 homodimer and that the stoichiometry of the ULK1–ATG13–FIP200 complex is 1:1:2 (*Alers et al., 2011*; *Ganley et al., 2009*; *Hieke et al., 2015*; *Hosokawa et al., 2009*; *Jung et al., 2009*; *Papinski and Kraft, 2016*; *Shi et al., 2020*; *Wallot-Hieke et al., 2018*). However, the structural details underlying the ULK complex assembly, particularly the role of the intrinsically disordered region of ATG13, remain elusive. To understand the mechanism of the ULK complex formation, we predicted the structure of the FIP200 [1–634 amino acids (aa)] dimer complexed with ATG13 (isoform c, 363–517 aa) and ULK1 (801–1050 aa) (underlined regions in *Figure 1A*) by AlphaFold2 with the AlphaFold-Multimer option, during which the appropriateness of the regions chosen for our predictions as interaction interfaces was further corroborated by AlphaFold2 models of full-length ATG13 with full-length ULK1 and of full-length ATG13 with the FIP200 (residues 1–634) dimer (*Figure 1—figure supplement 1*). *Figure 1B* depicts the predicted overall structure of the ULK1–ATG13–FIP200 complex, with flexible FIP200 loop regions that received low pLDDT scores omitted for clarity. The overall structure is consistent with the low-resolution cryo-EM model reported previously (*Shi et al., 2020*), including the C-shape of the FIP200 dimer and the binding of one molecule each of ATG13 and ULK1 to the FIP200 dimer, resulting in the 1:1:2 stoichiometry of the ULK1–ATG13–FIP200

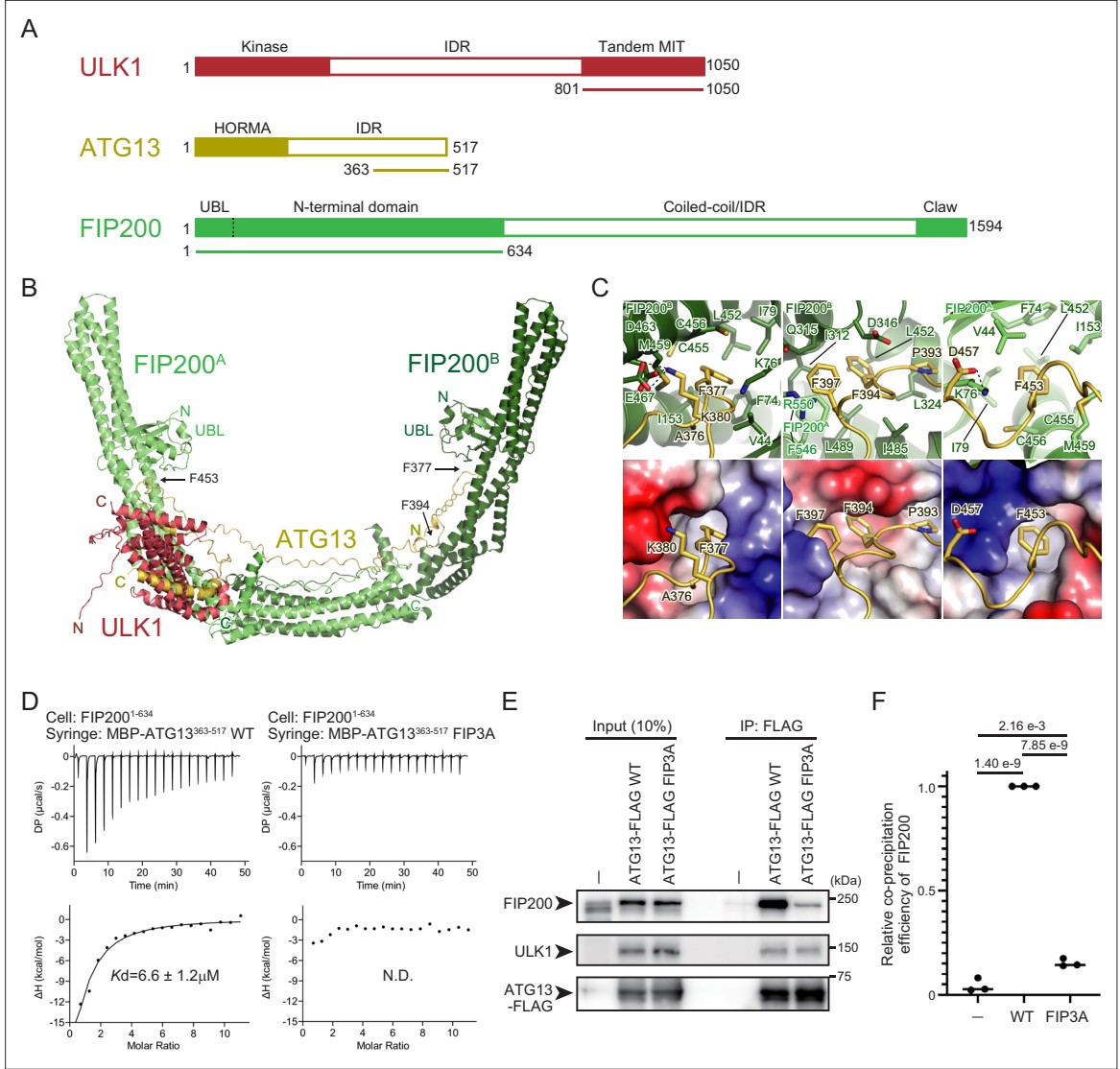

**Figure 1.** Structural basis of the ATG13–FIP200 interaction. (**A**) Domain architecture of ULK1, ATG13, and FIP200. Regions used for the AlphaFold2 complex prediction are underlined. (**B**) Structure of the ULK1–ATG13–FIP200 core complex predicted by AlphaFold2. Flexible loop regions in FIP200 were removed from the figure for clarity. N and C indicate N- and C-terminal regions, respectively. (**C**) Close-up view of the interactions between ATG13 and FIP200. The bottom panels represent the surface model of FIP200 with the coloring based on the electrostatic potentials (blue and red indicate positive and negative potentials, respectively). (**D**) Isothermal titration calorimetry (ITC) results obtained by titration of MBP-ATG13 (363–517 aa) WT or FIP3A mutant into an FIP200 (1–634 aa) solution. (**E**) Effect of the ATG13 FIP3A mutation on the FIP200 interaction in vivo. ATG13 KO HeLa cells stably expressing FLAG-tagged ATG13 WT or FIP3A were immunoprecipitated with an anti-FLAG antibody and detected with anti-FIP200, anti-ULK1, and anti-FLAG antibodies. (**F**) Relative amounts of precipitated FIP200 in (**E**) were calculated. Solid bars indicate the means, and dots indicate the data from three independent experiments. Differences were statistically analyzed using Tukey's multiple comparisons test.

The online version of this article includes the following source data and figure supplement(s) for figure 1:

**Source data 1.** PDF file containing original western blots for *Figure 1E*.

**Source data 2.** Original files for western blot analysis displayed in *Figure 1E*.

**Source data 3.** Values used for preparation of the graph in *Figure 1F*.

**Figure supplement 1.** AlphaFold2 prediction of the ULK1–ATG13–FIP200 core complex.

**Figure supplement 2.** Purification of recombinant proteins.

**Figure supplement 2—source data 1.** PDF file containing original western blots for *Figure 1—figure supplement 2*.

**Figure supplement 2—source data 2.** Original file for SDS–PAGE analysis displayed in *Figure 1—figure supplement 2*.

complex. The two FIP200 molecules are named FIP200[A] and FIP200[B], where FIP200[A] is the one close to ULK1. Extensive interactions were predicted between the four molecules, with a buried surface area of 3658 Å$^2$ for FIP200[A]–FIP200[B], 2195 Å$^2$ for FIP200[A]–ATG13, 1966 Å$^2$ for FIP200[B]–ATG13, 2140 Å$^2$ for ATG13–ULK1, and 1681 Å$^2$ for FIP200[A]–ULK1. No interaction was observed between FIP200[B] and ULK1. One ATG13 molecule binds to the FIP200 dimer through the interaction of residues 365–398 and 394–482 with FIP200[B] and FIP200[A], respectively, resulting in a 1:2 stoichiometry of the ATG13–FIP200 complex.

We compared our AlphaFold2 prediction model with the cryo-EM structures reported as a preprint recently (*Chen et al., 2023*). For the ULK1–ATG13–FIP200 complex, PDB entries 8SOI, 8SRM, and 8SQZ have global resolutions of 4.2, 4.46, and 5.85 Å, respectively, and employ different ATG13 segments (residues 462–517, 450–517, and 363–517) (*Figure 1—figure supplement 1F*). Despite these differences, all three cryo-EM models contain only the MIM region of ATG13 (residues 462–517). The ULK1 and FIP200 regions used also vary slightly, but are essentially the same as those we used for our AlphaFold2 calculations. In every structure, FIP200 forms a C-shaped dimer whose overall architecture resembles our AlphaFold2 model. While the C-shape opening angle differs among the structures, this variation is consistent with reports that FIP200 can adopt multiple conformations (*Shi et al., 2020*). The mode of ULK1 binding to FIP200 is likewise fundamentally conserved (a detailed comparison is given below). The main difference among the three structures is their stoichiometry: ULK1:ATG13:FIP200 is 1:1:2 in 8SOI, whereas it is 2:2:2 in the other two entries. In our AlphaFold2 model, ATG13 residues 365–390 bind one FIP200 protomer, and residues 442–472 bind the equivalent region of the second protomer, accounting for an ATG13:FIP200 stoichiometry of 1:2. Consequently, a 2:2 stoichiometry is not inconsistent when only the MIM segment of ATG13 is present. Conversely, PDB 8SQZ uses the same ATG13 segment (residues 363–517) as our AlphaFold2 calculations yet displays an ATG13:FIP200 stoichiometry of 2:2. Because this stoichiometry is observed only after the complex has been mixed with PI3KC3-C1 (*Chen et al., 2023*), some additional regulatory mechanism by PI3KC1-C1 is presumably involved, although its details remain unclear.

Previous studies using hydrogen–deuterium exchange coupled to mass spectrometry identified three regions (M1–M3) in ATG13 responsible for FIP200 binding (*Shi et al., 2020*). Among these sites, the side chain of Phe377 in M1 and Phe453 in M3 was deeply inserted into the hydrophobic pockets formed at the base of UBL domains (*Figure 1C*). Direct interaction between FIP200 (1–634) and Atg13 (363–517) was confirmed by isothermal titration calorimetry (ITC) with a $K_D$ value of 6.6 µM, which was severely attenuated by alanine substitution at these three phenylalanine sites (F377A, F394A, F453A; FIP3A) (*Figure 1D*, *Figure 1—figure supplement 2*). Furthermore, the ATG13[FIP3A] mutant expressed in *ATG13* KO cells displayed a significantly reduced coprecipitation rate of the endogenous FIP200 (*Figure 1E, F*). These results confirm that ATG13 and FIP200 interact with each other via the residues predicted by AlphaFold2 in vitro and in vivo.

## ULK1 and ATG13 interact via MIT–MIM interaction similar to yeast Atg1–Atg13

Next, we focused on the interaction between ULK1 and ATG13. We previously determined the crystal structure of the yeast Atg1–Atg13 complex, which revealed that the tandem microtubule-interacting and transport (MIT) domains in Atg1 bind to the tandem MIT-interacting motifs (MIMs) of ATG13 (*Figure 2A*, right; *Fujioka et al., 2014*). The predicted structure of the ULK1–ATG13 complex (*Figure 2A*, left) was quite similar to that of the yeast Atg1–Atg13 complex and to the cryo-EM structures of the ULK1–ATG13 portion of the ULK1 core complex (*Figure 2A*, middle). The assembly is stabilized by two sets of the ULK1[MIT1]–ATG13[MIM(C)] and ULK1[MIT2]–ATG13[MIM(N)] interactions (*Figure 2A*). Phe470 of ATG13[MIM(N)] and Phe512 of ATG13[MIM(C)] were inserted into the hydrophobic pocket of ULK1[MIT2] and ULK1[MIT1], respectively (*Figure 2B*). Direct interaction between ULK1 (636–1050) and Atg13 (363–517) was confirmed by ITC with a $K_D$ value of 0.34 µM, which was severely attenuated by alanine substitution at these two phenylalanine sites (F470A, F512A; ULK2A) (*Figure 2C*, *Figure 1—figure supplement 2*). In *ATG13* KO cells, the ULK1 expression level was significantly reduced, suggesting that the lack of ATG13 severely destabilized ULK1. This mirrors the ATG13-mediated stabilization of ATG101, another component of the ULK complex (*Suzuki et al., 2015*). The exogenous expression of ATG13[WT], but not of the ATG13[ULK2A] mutant, rescued the endogenous ULK1 expression (*Figure 2D, E*), indicating that the ULK2A mutation in ATG13 attenuates the ATG13–ULK1 interaction

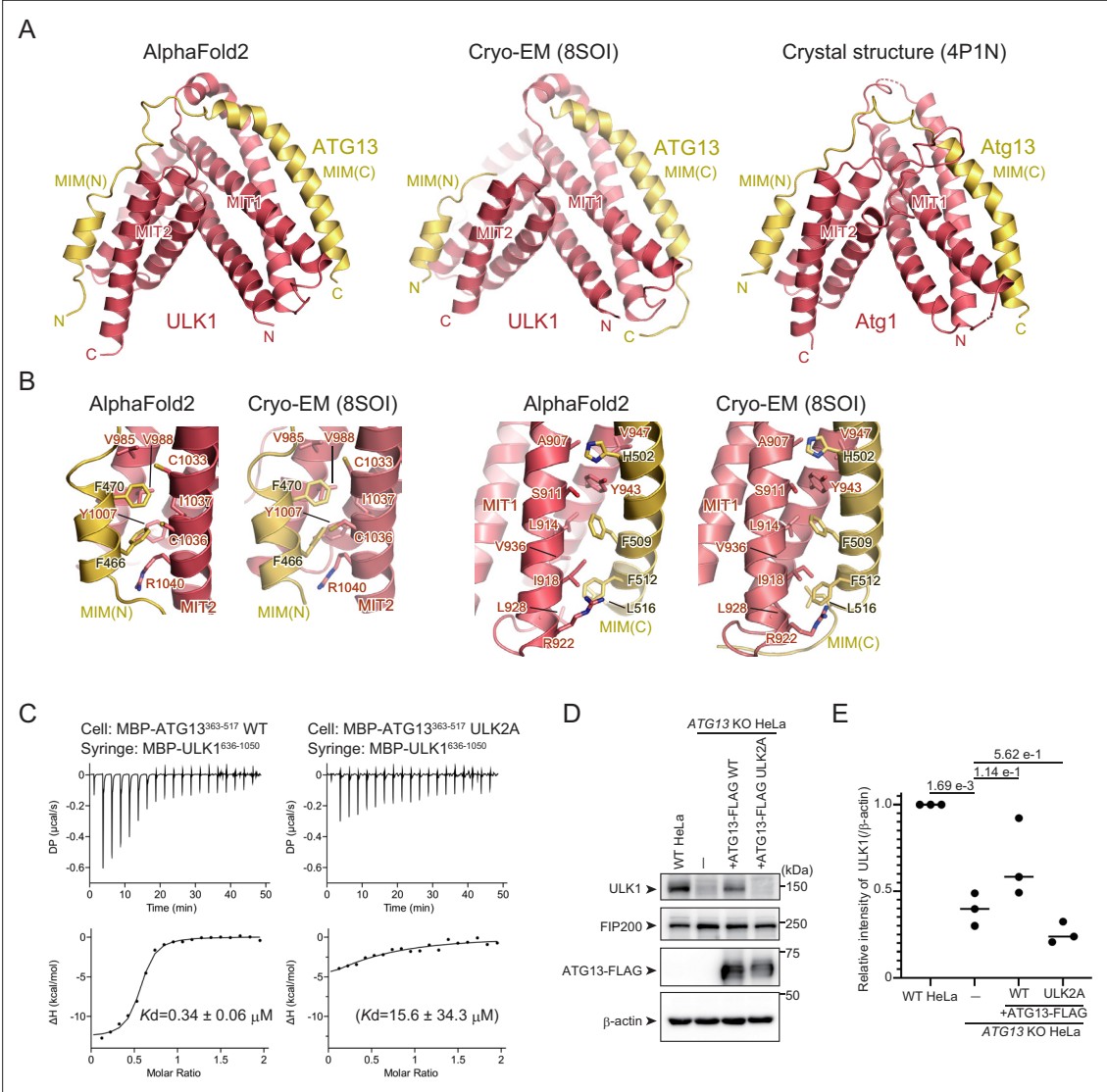

**Figure 2.** Structural basis of the ULK1–ATG13 interaction. (**A**) AlphaFold2 model of the ULK1–ATG13 moiety of the ULK1–ATG13–FIP200 core complex in *Figure 1B* (left), Cryo-EM structure of the ULK1–ATG13 moiety of the ULK1–ATG13–FIP200 core complex (PDB 8SOI), and crystal structure of the yeast Atg1–Atg13 complex (right, PDB 4P1N). (**B**) Close-up view of the interactions between ATG13$^{MIM(N)}$ and ULK1$^{MIT2}$ and between ATG13$^{MIM(C)}$ and ULK1$^{MIT1}$ (right). (**C**) Isothermal titration calorimetry (ITC) results obtained by titration of MBP-ULK1 (636–1050 aa) into a solution of WT or ULK2A mutant of MBP-ATG13 (363–517 aa). Due to weak binding, the $K_D$ value for the ULK2A mutant was not accurately determined. (**D**) Effect of the ATG13–FIP3A mutation on endogenous ULK1 levels in vivo. WT or *ATG13* KO HeLa cells stably expressing FLAG-tagged ATG13 WT or ULK2A mutant were lysed, and indicated proteins were detected by immunoblotting using anti-FIP200, anti-ULK1, and anti-FLAG antibodies. (**E**) Relative amounts of ULK1 in (**D**) were normalized with β-actin and calculated. Solid bars indicate the means, and dots indicate the data from three independent experiments. Differences were statistically analyzed using Tukey's multiple comparisons test.

The online version of this article includes the following source data for figure 2:

**Source data 1.** PDF file containing original western blots for *Figure 2D*.

**Source data 2.** Original files for western blot analysis displayed in *Figure 2D*.

**Source data 3.** Values used for preparation of the graph in *Figure 2E*.

in vivo. These observations confirm that, like the yeast Atg1–Atg13 interaction, the ULK1–ATG13 interaction is mediated by MIT–MIM contacts.

## Direct interaction between ULK1 and FIP200 is essential for autophagy

Although AlphaFold2 predicted the direct interaction between ULK1 and FIP200 over a reasonably large contact area (1681 Å$^2$) and similar interaction was observed in cryo-EM structures (*Figure 3A, B*; *Chen et al., 2023*), a previous study reported that ULK1 required ATG13–ATG101 for FIP200 binding in vitro (*Shi et al., 2020*). ULK1$^{MIT1}$ and ULK1$^{MIT2}$ interact directly with FIP200, with the Leu967 of ULK1$^{MIT1}$ and Phe997 of ULK1$^{MIT2}$ inserted on the hydrophobic groove of FIP200 (*Figure 3A, B*). Although we failed to measure the affinity between ULK1 and FIP200 by ITC due to aggregation upon their mixing, an in vitro pull-down assay indicated that ULK1 directly interacted with FIP200, which was significantly attenuated by the FIP2A (L967A, F997A) mutation (*Figure 3C, D*). Consistently, the ULK1$^{FIP2A}$ mutant lost the interaction with FIP200 while partially retaining the interaction with ATG13 in cells (*Figure 3E, F*). We confirmed that the ULK1$^{FIP2A}$ mutant retains substantial affinity with ATG13 in vitro by ITC (*Figure 3—figure supplement 1*). These results confirm the direct ULK1–FIP200 interaction in vitro and in vivo. Our results establish the mode of the three independent interactions within the ULK complex, ATG13–FIP200, ATG13–ULK1, and ULK1–FIP200, operating both in vitro and in vivo.

To investigate the physiological importance of the ULK1–FIP200 interaction, we assessed autophagy flux using the Halo-LC3 processing assay (*Rudinskiy et al., 2022*; *Yim et al., 2022*). *Ulk1,2* DKO mouse embryonic fibroblasts (MEFs) expressing FLAG-tagged wild-type ULK1 produce a stronger band of cleaved Halo than cells which do not express ULK1. The intensity of this cleaved Halo band corresponds to the amount of Halo-LC3 delivered to lysosomes in an autophagy-dependent manner. *Ulk1,2* DKO MEFs expressing the ULK1$^{FIP2A}$ mutant had partially but significantly reduced processed Halo bands upon starvation compared to ULK1$^{WT}$-expressing cells, confirming that the direct ULK1–FIP200 interaction is crucial, although partially, for autophagy activity (*Figure 3G, H*). This partial phenotype would be due to an indirect interaction between ULK1 and FIP200 via ATG13, which was supported by the observation that the ULK1$^{FIP2A}$ mutant colocalized with FIP200 in cells, although less efficiently than ULK1$^{WT}$ (*Figure 3I*).

## Triadic ULK1–ATG13–FIP200 interactions are critical for autophagic flux

We compared the autophagic flux of impaired ATG13–ULK1 and ATG13–FIP200 interactions to confirm the hypothesis that indirect triadic complexes are functional for autophagy. We generated knock-in (KI) cell lines that included *ATG13$^{FIP3A}$*, *ATG13$^{ULK2A}$*, and *ATG13$^{FU5A}$* (F377A, F394A, F453A, F470A, F512A; FU5A) followed by a 3xFLAG-tag in the genomic *ATG13* locus of HeLa cells (*Figure 4A, B*). Since overexpressed ATG13-FLAG is more than 20-fold higher than its endogenous level, the mutation significance can be evaluated at more physiological expression levels using the KI cell lines (*Figure 4—figure supplement 1A, B*). The ATG13$^{FIP3A}$ and ATG13$^{ULK2A}$ KI cells displayed partial colocalization with FIP200, whereas the ATG13$^{FU5A}$ KI cells displayed scarce colocalization with FIP200 (*Figure 4C*). Next, we expressed Halo-LC3 in these KI cells and performed a Halo processing assay. Cleaved Halo was partially reduced in ATG13$^{FIP3A}$ and ATG13$^{ULK2A}$ KI cells compared to ATG13$^{WT}$ cells. In contrast, cleaved Halo in ATG13$^{FU5A}$ KI cells was almost equivalent to that in the knockout cells (*Figure 4D, E*). Note that ATG13$^{WT}$ KI cells show a ~25% reduction in cleaved Halo compared to WT HeLa cells, suggesting that FLAG-tag KI only partially affects autophagic flux. These results suggest that ATG13–ULK1 and ATG13–FIP200 bindings complement each other in autophagy function and that ULK1, ATG13, and FIP200 directly bind to each other to organize the robust ULK complex.

Finally, we investigated how disrupting the ULK complex leads to impaired autophagy. One well-established function of the ULK complex is to recruit ATG9 vesicles (*Ren et al., 2023*). These vesicles serve as an upstream platform for the PI3KC3-C1, providing the substrate for phosphoinositide generation (*Holzer et al., 2024*). To clarify how our mutations impact this step, we starved ATG13 mutant KI cells and observed ATG9A localization. Unexpectedly, even in ATG13$^{FU5A}$ KI cells, where ATG13 is almost completely dissociated from the ULK complex, ATG9A still colocalized with FIP200 (*Figure 4—figure supplement 1C*). Since these puncta also overlapped with p62, it is likely that p62 bodies recruit both FIP200 and ATG9 vesicles (*Hama et al., 2023*). Therefore, it is apparently difficult to assess the autophagosome formation-associated recruitment of the ATG9 vesicles using the present

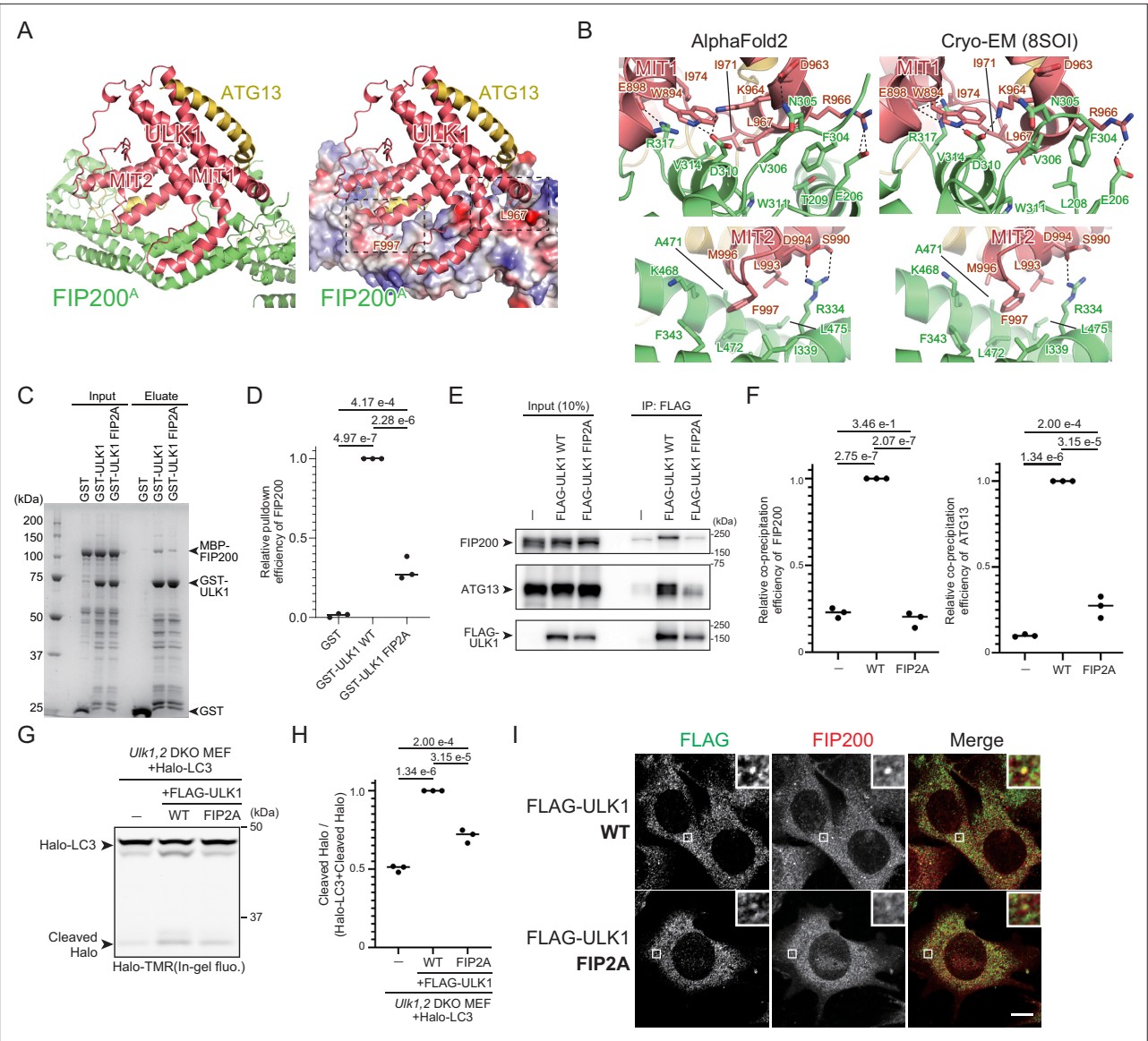

**Figure 3.** Structural basis of the ULK1–FIP200 interaction. (**A**) Structure of the ULK1–FIP200 moiety of the ULK1–ATG13–FIP200 core complex in *Figure 1B*. The right panel represents the surface model of FIP200 with coloring based on the electrostatic potentials (blue and red indicate positive and negative potentials, respectively). Dotted squares indicate the regions displayed in (**B**). (**B**) Close-up view of the interactions between ULK1$^{MIT1}$ and FIP200 (top) and between ULK1$^{MIT2}$ and FIP200 (bottom). Left and right indicate AlphaFold2 and cryo-EM (PDB 8SOI) models. (**C**) In vitro pull-down assay between GST-ULK1 (636–1050 aa) WT or FIP2A mutant with MBP-FIP200 (1–634 aa). (**D**) Relative amounts of precipitated MBP-FIP200 in (**C**) were calculated. Solid bars indicate the means, and dots indicate the data from three independent experiments. Differences were statistically analyzed using Tukey's multiple comparisons test. (**E**) Effect of the ULK1 FIP2A mutation on the FIP200 interaction in vivo. Ulk1,2 DKO mouse embryonic fibroblasts (MEFs) stably expressing FLAG-tagged ULK1 WT or FIP2A mutant were immunoprecipitated with an anti-FLAG antibody and detected with anti-FIP200, anti-ATG13, and anti-FLAG antibodies. (**F**) Relative amounts of precipitated FIP200 (left) and ATG13 (right) in (**E**) were calculated. Solid bars indicate the means, and dots indicate the data from three independent experiments. Differences were statistically analyzed using Tukey's multiple comparisons test. (**G**) Halo-LC3 processing assay of ULK1 FIP2A-expressing cells. *Ulk1,2* DKO MEFs stably expressing Halo-LC3 and FLAG-tagged ULK1 WT or FIP2A mutant were labeled for 15 min with 100 nm tetramethylrhodamine (TMR)-conjugated Halo ligand and incubated in starvation medium for 1 hr. Cell lysates were subjected to in-gel fluorescence detection. (**H**) Halo processing rate in (**G**). The band intensity of processed Halo and Halo-LC3 in each cell line was quantified, and the relative cleavage rate was calculated as FLAG-ULK1 WT-expressing cells as 1. Solid bars indicate the means, and dots indicate the data from three independent experiments. Data were statistically analyzed using Tukey's multiple comparisons test. (**I**) Colocalization of FLAG-ULK1 WT or FIP2A mutant with FIP200. *Ulk1,2* DKO MEFs stably expressing FLAG-tagged ULK1 WT or FIP2A mutant were immunostained with anti-FLAG and anti-FIP200 antibodies. Scale bar, 10 µm.

The online version of this article includes the following source data and figure supplement(s) for figure 3:

*Figure 3 continued on next page*

*Figure 3 continued*

**Source data 1.** PDF file containing original western blots or SDS–PAGE for *Figure 3C, E, G*.

**Source data 2.** Original files for western blot or SDS–PAGE analysis displayed in *Figure 3C, E, G*.

**Source data 3.** Values used for preparation of the graph in *Figure 3D, F, H*.

**Figure supplement 1.** Isothermal titration calorimetry (ITC) analysis between ULK1 and ATG13.

KI cell lines. Another key function of the ULK complex is the ULK1-dependent phosphorylation of ATGs, and phosphorylation of ATG14 at Ser29 is known to be particularly critical (*Park et al., 2016*; *Wold et al., 2016*). In ATG13^FIP3A and ATG13^FU5A KI cells, ATG14 phosphorylation was significantly reduced, indicating decreased ULK1 activity (*Figure 4—figure supplement 1D, E*). Notably, in *ATG13* KO cells, ATG14 phosphorylation became almost undetectable, though the underlying mechanism remains to be fully investigated. Altogether, these data point to reduced ULK1 activity as a key factor explaining the autophagy deficiency observed in ATG13^FU5A KI cells.

## Discussion

The interaction mechanism of ULK1–ATG13–FIP200, the core of the ULK complex, has been extensively studied (*Alers et al., 2011*; *Chen et al., 2023*; *Ganley et al., 2009*; *Hieke et al., 2015*; *Hosokawa et al., 2009*; *Jung et al., 2009*; *Papinski and Kraft, 2016*; *Shi et al., 2020*). However, most of these studies either lacked structural context or, when structural analyses were performed, they were not adequately validated by functional assays in both in vitro and in vivo settings, leaving our understanding of the interaction mechanism still limited. In this study, AlphaFold2-based point mutational analysis provides solid evidence for the direct ULK1–ATG13, ATG13–FIP200, and ULK1–FIP200 bindings in vitro and in vivo. Additionally, the triadic interaction is complementary in the cell.

Regarding the importance of triadic interactions for the molecular function of the ULK complex, we observed their partial significance for ULK1 kinase activity against ATG14, a component of PI3KC3-C1, in our assays (*Figure 4—figure supplement 1D, E*). This observation is consistent with recent work showing that FIP200 interacts with the PI3KC3-C1 (*Chen et al., 2023*). In contrast, we were unable to evaluate autophagosome formation-associated recruitment of the ATG9 vesicles in our ATG13 KI cell lines (*Figure 4—figure supplement 1C*). However, given the direct interaction between ATG13 and ATG9A, it is likely that recruitment fails in ATG13^FU5A KI cells. In the future, recruitment should be reevaluated under conditions that prevent p62 body-mediated recruitment, such as a penta-receptor KO genetic background.

One major structural difference between the mammalian ULK complex we predicted here and the yeast Atg1 complex is that one ATG13 binds to two FIP200s within the same FIP200 dimer in mammals, whereas one Atg13 binds to two Atg17s in the distinct Atg17 dimers in yeast (*Figure 4F*). The latter binding mode of Atg13 bridges Atg17 dimers to each other to form a higher-order assemblage (*Yamamoto et al., 2016*), which is considered to be the Atg1 complex's phase separation mechanism (*Fujioka et al., 2020*). Conversely, ATG13 cannot bridge FIP200 dimers to each other and thus cannot induce a higher-order assemblage of the ULK complex, which necessitates another mechanism for phase separation. A recent study reported that calcium transients on the endoplasmic reticulum surface trigger FIP200 phase separation (*Zheng et al., 2022*); however, the detailed mechanisms are unresolved. Further studies are required to elucidate the mechanisms of phase separation in organizing the autophagosome formation sites in mammals.

In addition to Atg17, budding and fission yeasts have Atg11 as a closer homolog of FIP200 (*Kim et al., 2001*; *Li et al., 2014*; *Pan et al., 2020*; *Sun et al., 2013*; *Yorimitsu and Klionsky, 2005*). Atg11 and Atg17 interact with Atg1 and Atg13, respectively, whereas Atg11–Atg13 and Atg17–Atg1 interactions have not been reported. *Arabidopsis thaliana*, which belongs to the Archaeplastida group, a supergroup distant from Opisthokonta (including yeasts and mammals), also has Atg1, Atg13, and Atg11 (*Burki et al., 2020*; *Li et al., 2014*). The *A. thaliana* Atg11 interacts with Atg13 but not with Atg1 (*Li et al., 2014*). The triadic interaction of Atg1/ULK1, ATG13, and Atg11/Atg17/FIP200 has not been reported for any species other than mammals. This triadic interaction is likely beneficial for cells that form the complicated cellular communities that comprise the metazoans. The most likely physiological significance is the fine-tuning of tissue-specific autophagic activity. Consistent with this

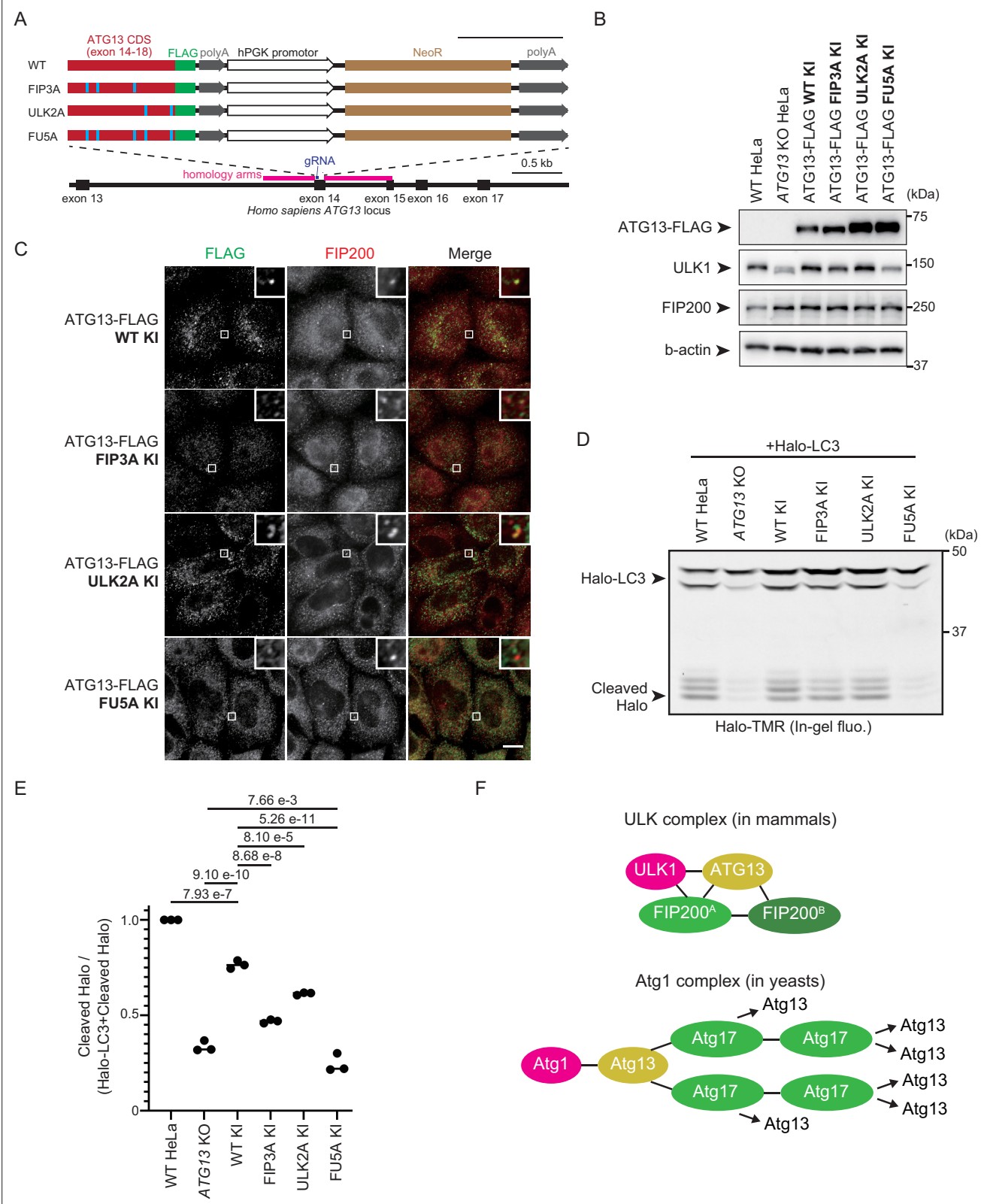

**Figure 4.** ATG13–ULK1 and ATG13–FIP200 interactions complement each other in autophagy function. (**A**) Schematic representation of the CRISPR–Cas9-mediated KI strategy of ATG13 mutations with FLAG tag. The C-terminally FLAG-tagged coding sequence after exon 14 of ATG13 with or without FIP3A, ULK2A, or FU5A mutations were knocked in exon 14 of the *Homo sapiens* ATG13 locus. As the KI cassette expresses NeoR under the hPGK1 promoter, clones that were successfully knocked in were selected by G418. Cas9-gRNA-targeted sites in the exon 14 of *H. sapiens* ATG13 locus are

*Figure 4 continued on next page*

*Figure 4 continued*

displayed in dark blue. The homology arm for KI is presented in magenta, and the ATG13 CDS and mutations in red and cyan, respectively. NeoR is displayed in brown. Scale bar, 0.5 kilobase pair (kb). (**B**) Immunoblot of ATG13-FLAG KI cell lines. WT, *ATG13* KO, and indicated KI HeLa cells were lysed, and indicated proteins were detected by immunoblotting using anti-FIP200, anti-ULK1, and anti-FLAG antibodies. (**C**) Colocalization of endogenous levels of ATG13-FLAG mutants with FIP200. Indicated KI cell lines were cultured in the starvation medium for 1 hr and immunostained with anti-FLAG and anti-FIP200 antibodies. Scale bar, 10 μm. (**D**) Halo-LC3 processing assay of ATG13-FLAG KI cell lines. WT, ATG13 KO and KI HeLa cell lines were labeled for 15 min with 100 nm tetramethylrhodamine (TMR)-conjugated Halo ligand and incubated in starvation medium for 1 hr. Cell lysates were subjected to in-gel fluorescence detection. (**E**) Halo processing rate in (**D**). The band intensity of processed Halo and Halo-LC3 in each cell line was quantified, and the relative cleavage rate was calculated as WT HeLa cells as 1. Solid bars indicate the means, and dots indicate the data from three independent experiments. Data were statistically analyzed using Tukey's multiple comparisons test. (**F**) Schematic depiction of the difference between the mammalian ULK complex and the yeast Atg1 complex. Mammalian ATG13 binds to two FIP200s within the same FIP200 dimer, contributing to the stability of one ULK complex. Conversely, budding yeast Atg13 binds to two Atg17s within a different Atg17 dimer, allowing for endlessly repeated Atg13–Atg17 interactions. ATG101 in the ULK complex and Atg31-29 in the Atg1 complex are omitted for simplicity. ATG13/Atg13 is shown in yellow, ULK1/Atg1 in magenta, and FIP200/Atg17 in green. Black lines represent interactions.

The online version of this article includes the following source data and figure supplement(s) for figure 4:

**Source data 1.** PDF file containing original western blots for *Figure 4B, D*.

**Source data 2.** Original files for western blot analysis displayed in *Figure 4B, D*.

**Source data 3.** Values used for preparation of the graph in *Figure 4E*.

**Figure supplement 1.** Stable overexpression of *ATG13* results in 10-fold higher protein expression than the endogenous version.

**Figure supplement 1—source data 1.** PDF file containing original western blots for *Figure 4—figure supplement 1A, D*.

**Figure supplement 1—source data 2.** Original files for western blot analysis displayed in *Figure 4—figure supplement 1A, D*.

**Figure supplement 1—source data 3.** Values used for preparation of the graph in *Figure 4—figure supplement 1B, E*.

idea, several splicing variants of ATG13 are present in vertebrates (*Alers et al., 2014*; *Hieke et al., 2015*; *Jung et al., 2009*). In humans, at least five splicing variants are known, including the isoform 3 lacking the C-terminal ULK1-binding site (MIMs, *Figure 2A*; *Alers et al., 2014*; *Jung et al., 2009*). The avian Atg13 has seven splicing variants, some lacking parts of the FIP200 interaction site (*Alers et al., 2014*; *Hieke et al., 2015*). This study found that a single disruption of the FIP200–ATG13 or ULK1–ATG13 interaction results in a partial reduction of autophagy activity (*Figure 4D*). In the individual context, the tissue- and organ-specific fine-tuning of the autophagy activity may occur by regulating the expression amount and ratio of splicing variants that lack some of the triad interactions. Generating KI mice with the point mutations identified in this study might help investigate the significance of the triadic binding at the individual level.

# Materials and methods

### Key resources table

| Reagent type (species) or resource | Designation | Source or reference | Identifiers | Additional information |
|---|---|---|---|---|
| Gene (*Homo sapiens*) ULK1 | | NCBI Reference Sequence | NM_003565.4 | |
| Gene (*H. sapiens*) | ATG13 (isoform c) | *Hosokawa et al., 2009* | NM_001142673.3 | |
| Gene (*H. sapiens*) | RB1CC1/FIP200 (isoform 2) | *Hara et al., 2008* | NM_001083617.2 | |
| Gene (*H. sapiens*) | ATG9A | *Kishi-Itakura et al., 2014* | NM_001077198.3 | |
| Gene (*Rattus norvegicus*) | MAP1LC3B | *Yim et al., 2022* | NM_022867.2 | |
| Strain (*Escherichia coli*) | BL21 (DE3) | Novagen | 69450 | |
| Cell line (*H. sapiens*) | HEK293T | RIKEN | RCB2202 | |
| Cell line (*H. sapiens*) | HeLa | RIKEN | RCB0007 | |
| Cell line (*M. musculus*) | *Ulk1 Ulk2* DKO MEF | *Cheong et al., 2011* | Kindly provided by Craig B, Thompson | Established from C57BL/6 mice |

*Continued on next page*

*Continued*

| Reagent type (species) or resource | Designation | Source or reference | Identifiers | Additional information |
|---|---|---|---|---|
| Cell line (*H. sapiens*) | *ATG13* KO HeLa | **Hama et al., 2023** | WM1 | |
| Cell line (*H. sapiens*) | ATG13-3xFLAG KI HeLa | This study | 13FWTKI14 | |
| Cell line (*H. sapiens*) | ATG13-3xFLAG FIP3A KI HeLa | This study | 13FF3AKI12 | |
| Cell line (*H. sapiens*) | ATG13-3xFLAG ULK2A KI HeLa | This study | 13FU2AKI12 | |
| Cell line (*H. sapiens*) | ATG13-3xFLAG FU5A KI HeLa | This study | 13FFU5AKI24 | |
| Antibody | Rabbit polyclonal anti-FIP200 | Proteintech | 17250-1-AP | 1:1000 for WB, 1:500 for IF |
| Antibody | Rabbit monoclonal anti-ATG13 | Cell Signaling Technology | #13273 | 1:1000 for WB |
| Antibody | Rabbit polyclonal anti-ULK1 | Cell Signaling Technology | #8054S | 1:500 for WB |
| Antibody | Rabbit polyclonal anti-ATG14 p-Ser29 | Cell Signaling Technology | #92340 | 1:1000 for WB |
| Antibody | Rabbit polyclonal anti-ATG14 | Proteintech | 24412-1-AP | 1:1000 for WB |
| Antibody | Guinea pig polyclonal anti-p62 | PROGEN | GP62-C | 1:500 for IF |
| Antibody | Mouse monoclonal anti-FLAG | MBL | M185-7 | 1:1000 for WB, 1:500 for IF |
| Antibody | Mouse monoclonal anti-HA | MBL | M180-3 | 1:500 for IF |
| Antibody | Mouse monoclonal anti-β-actin | Sigma-Aldrich | A2228 | 1:10,000 for WB |
| Antibody | HRP-conjugated mouse monoclonal anti-DDDDK tag | MBL | M185-7 | 1:2000 for WB |
| Antibody | HRP-conjugated mouse monoclonal anti-rabbit IgG | Jackson ImmunoResearch | 111-035-144 | 1:10,000 for WB |
| Antibody | HRP-conjugated mouse monoclonal anti-mouse IgG | Jackson ImmunoResearch | 111-035-003 | 1:10,000 for WB |
| Antibody | Alexa Fluor 488-conjugated polyclonal anti-mouse IgG | Thermo Fisher Scientific | A-11029 | 1:2000 for IF |
| Antibody | Alexa Fluor 555-conjugated polyclonal anti-rabbit IgG | Thermo Fisher Scientific | A-31572 | 1:2000 for IF |
| Antibody | Alexa Fluor 647-conjugated polyclonal anti-pig IgG | Thermo Fisher Scientific | A-21450 | 1:2000 for IF |
| Recombinant DNA reagent | pET15b | Novagen | 69661 | |
| Recombinant DNA reagent | pGEX-6P-1 | Cytiva | 28954648 | **Figure 3C** |
| Recombinant DNA reagent | pET15b-MBP-ULK1 (636–1050 aa) | This study | | **Figure 2C** |
| Recombinant DNA reagent | pET15b-MBP-ULK1$^{FIP2A}$ (636–1050 aa) | This study | | **Figure 3—figure supplement 1** |
| Recombinant DNA reagent | pET15b-MBP-ATG13 (363–517 aa) | This study | | **Figures 1D, 2C, Figure 1—figure supplement 2** |
| Recombinant DNA reagent | pET15b-MBP-ATG13$^{FIP3A}$ (363–517 aa) | This study | | **Figure 1D** |
| Recombinant DNA reagent | pET15b-MBP-ATG13$^{ULK2A}$ (363–517 aa) | This study | | **Figure 2C** |
| Recombinant DNA reagent | pET15b-MBP-FIP200 (1–634 aa) | This study | | **Figures 1D and 3C** |

*Continued on next page*

*Continued*

| Reagent type (species) or resource | Designation | Source or reference | Identifiers | Additional information |
|---|---|---|---|---|
| Recombinant DNA reagent | pGEX6p-1-GST-ULK1 (636–1050 aa) | This study | | *Figure 3C* |
| Recombinant DNA reagent | pGEX6p-1-GST-ULK1FIP2A (636–1050 aa) | This study | | *Figure 3C* |
| Recombinant DNA reagent | pMRX-IP-3xFLAG-ULK1 | This study | YHE134 | *Figure 3E, G* |
| Recombinant DNA reagent | pMRX-IP-3xFLAG-ULK1FIP2A | This study | YHE141 | *Figure 3E, G* |
| Recombinant DNA reagent | pMRX-IP-ATG13-3xFLAG | This study | YHE103 | *Figures 1D, 2D, Figure 4—figure supplement 1A* |
| Recombinant DNA reagent | pMRX-IP-ATG13FIP3A-3xFLAG | This study | YHE116 | *Figure 1D* |
| Recombinant DNA reagent | pMRX-IP-ATG13ULK2A-3xFLAG | This study | YHE144 | *Figure 2D* |
| Recombinant DNA reagent | pMRX-IP-ATG13FU5A-3xFLAG | This study | YHE180 | *Figure 4* |
| Recombinant DNA reagent | pMRX-IP-ATG9A-3xHA | This study | YHE218 | *Figure 4—figure supplement 1C* |
| Recombinant DNA reagent | pCG-gag-pol | Kindly provided by Teruhiko Yasui | | For packaging retrovirus for stable gene expression |
| Recombinant DNA reagent | pCG-VSV-G | Kindly provided by Teruhiko Yasui | | For packaging retrovirus for stable gene expression |
| Recombinant DNA reagent | PX458-gATG13 exon14 | This study | YHC49 | *Figure 4* |
| Recombinant DNA reagent | pKnockIn | This study | YHC3 | *Figure 4* |
| Recombinant DNA reagent | pKnockIn-ATG13 exon14-ATG13 (347–517 aa)–3xFLAG | This study | YHC50 | *Figure 4* |
| Recombinant DNA reagent | pKnockIn-ATG13 exon14-ATG13FIP3A (347–517 aa)–3xFLAG | This study | YHC51 | *Figure 4* |
| Recombinant DNA reagent | pKnockIn-ATG13 exon14-ATG13ULK2A (347–517 aa)–3xFLAG | This study | YHC52 | *Figure 4* |
| Recombinant DNA reagent | pKnockIn-ATG13 exon14-ATG13FU5A (347–517 aa)–3xFLAG | This study | YHC53 | *Figure 4* |
| Peptide, recombinant protein | PrimeSTAR Max DNA Polymerase | Takara Bio | R045A | |
| Peptide, recombinant protein | NEBuilder HiFi DNA Assembly Master Mix | New England Biolabs | E2621X | |
| Chemical compound, drug | FuGENE HD | Promega | VPE2311 | |
| Chemical compound, drug | HaloTag TMR Ligand | Promega | G8251 | |
| Chemical compound, drug | Digitonin | Sigma-Aldrich | D141 | |
| Chemical compound, drug | Polybrane | Sigma-Aldrich | H9268 | |
| Chemical compound, drug | Puromycin | Sigma-Aldrich | P8833 | |

*Continued on next page*

*Continued*

| Reagent type (species) or resource | Designation | Source or reference | Identifiers | Additional information |
|---|---|---|---|---|
| Chemical compound, drug | Blasticidin | Fujifilm Wako Pure Chemical Corporation | 022-18713 | |
| Chemical compound, drug | G-418 | Fujifilm Wako Pure Chemical Corporation | 074-06801 | |
| Chemical compound, drug | Anti-FLAG M2 affinity gel | Sigma-Aldrich | A2220 | |
| Chemical compound, drug | 4% paraformaldehyde | Fujifilm Wako Pure Chemical Corporation | 163-20145 | |
| Chemical compound, drug | Amylose Resin High Flow | New England Biolabs | E8022L | |
| Chemical compound, drug | Bio-Scale Mini Bio-Gel P-6 desalting column | Bio-Rad Laboratories | 7325304 | |
| Chemical compound, drug | COSMOGEL GST-Accept | NACALAI TESQUE | 09277-14 | |
| Chemical compound, drug | HiLoad 26/600 Superdex 200 pg | Cytiva | 28989336 | |
| Chemical compound, drug | One Step CBB | BIO CRAFT | CBB-1000 | |
| Software, algorithm | AlphaFold2 v2.3 | *Jumper et al., 2021* | | Structural prediction was done using AlphaFold2 v2.3 |
| Software, algorithm | MicroCal PEAQ-ITC analysis software | Malvern Panalytical Ltd | | Integration and fitting of ITC were done using MicroCal PEAQ-ITC analysis software |
| Software, algorithm | Fiji-ImageJ | https://imagej.net/Fiji/Downloads | | Image analysis was done using Fiji-ImageJ and plugins |
| Software, algorithm | Illustrator | Adobe | | Images were mounted using these softwares |
| Software, algorithm | GraphPad Prism 9 | GraphPad Prism | | Graphs and statistical tests were done using GraphPad Prism |

## Structural prediction using AlphaFold2 with the AlphaFold-Multimer mode

Structure was predicted using AlphaFold2 v2.3 installed on a local computer (Sunway Technology Co, Ltd) (*Jumper et al., 2021*). The predictions were run using the AlphaFold-Multimer mode (*Evans et al., 2021*), with five models and a single seed per model, and default multiple sequence alignment generation using the MMSeqs2 server (*Mirdita et al., 2022*; *Mirdita et al., 2019*). The unrelaxed predicted models were subjected to an Amber relaxation procedure, and the relaxed model with the highest confidence based on predicted LDDT scores was selected as the best model and used for figure preparation (*Jumper et al., 2021*). Structural figures were prepared using PyMOL (http://www.pymol.org/pymol). The model is available in ModelArchive (https://modelarchive.org/) with the accession code ma-jz53c.

## Cell culture

HeLa cells and MEFs were cultured in Dulbecco's modified Eagle medium (DMEM) (043-30085; Fujifilm Wako Pure Chemical Corporation) supplemented with 10% fetal bovine serum (FBS) (172012; Sigma-Aldrich) in a 5% $CO_2$ incubator at 37°C. Cells were washed twice with phosphate-buffered saline (PBS) and incubated in amino acid-free DMEM (048-33575; Fujifilm Wako Pure Chemical Corporation) without FBS for starvation. HeLa and HEK293T cells were authenticated and validated by STR profiling by RIKEN. *ATG13* KO HeLa cells (*Hama et al., 2023*) and *Ulk1 Ulk2* double knockout MEFs

(*Cheong et al., 2011*) were described previously. *Ulk1 Ulk2* double knockout MEFs were originally derived from C57BL/6 mouse embryos and provided by an external researcher. The mammalian cell lines were routinely confirmed to be free of mycoplasma contamination by fluorescence microscopy.

## Plasmids for mammalian cells

Plasmids for stable expression were generated as follows: DNA fragments encoding human ULK1 (NM_003565.4), ATG13 (*Hosokawa et al., 2009*), and ATG9A (*Itakura et al., 2012*) were inserted into the retroviral plasmid pMRX-IP (puromycin-resistant marker) (*Kitamura et al., 2003*; *Saitoh et al., 2003*) with a 3 × FLAG or 3 × HA epitope tag. Site-directed mutagenesis was used to introduce ATG13$^{FIP3A}$, ATG13$^{ULK2A}$, ATG13$^{FU5A}$, and ULK1$^{FIP2A}$ mutations. pMRXIB-HaloTag7-ratLC3B was described previously (*Yim et al., 2022*). Plasmids for *ATG13* KI were generated as follows: sgRNAs targeting the exon 14 of *ATG13* (5′-GGCCTCCCCTCACGATGTCT-3′) were inserted into the *Bpi*I site of PX458 [pSpCas9(BB)-2A-GFP; Addgene #48138]. Donor plasmids for *ATG13* KI were generated as follows: 500 base-pair (bp) and 682 bp homology arms were amplified from the genomic *ATG13* locus around exon 14 and inserted to flank the KI cassettes (described in *Figure 4A*) in the donor plasmid (pKnockIn). The coding sequence of 3xFLAG-tagged ATG13 with or without FIP3A, ULK2A, or FU5A mutations was amplified from stable expression plasmids (described above) and inserted just behind the 5′ homology arm.

## Stable expression in HeLa cells by retroviral infection

HEK293T cells were transfected with the retroviral plasmid with pCG-gag-pol and pCG-VSV-G (a gift from Dr. T. Yasui, National Institutes of Biomedical Innovation, Health and Nutrition) using FuGENE HD (E2311; Promega) for 4 hr in Opti-MEM (31985-070; Gibco). After cell cultivation for 2 days in DMEM, the retrovirus-containing medium was harvested, filtered through a 0.45-μm filter unit (Ultrafree-MC; Millipore), and added to HeLa cells or MEFs with 8 μg/ml polybrene (H9268; Sigma-Aldrich). After 24 hr, 2 μg/ml puromycin (P8833; Sigma-Aldrich) or 2.5 μg/ml blasticidin (022-18713; Fujifilm Wako Pure Chemical Corporation) was used to select the stable transformants.

## Plasmids for protein preparation

To construct the expression plasmid encoding an N-terminal maltose-binding protein (MBP) followed by an HRV3C protease site, the genes were amplified by PCR and cloned into the pET15b vector. The genes were amplified by PCR and cloned into the pET15b-MBP vector for the plasmids encoding N-terminal MBP-tagged ULK1, ATG13, and FIP200. Similarly, for the plasmids encoding N-terminal glutathione *S*-transferase (GST)-tagged ULK1, the genes were amplified by PCR and cloned into the pGEX6p-1 vector. The NEBuilder HiFi DNA Assembly Master Mix (New England BioLabs) was used to assemble the PCR fragments. PCR-mediated site-directed mutagenesis was used to introduce the mutations that led to the specified amino acid substitutions. All constructs were sequenced to confirm their identities.

## Protein expression and purification

*E. coli* BL21 (DE3) cells were used to express all recombinant proteins. After cell lysis, MBP-FIP200 (1–634) was purified by affinity chromatography using Amylose Resin High Flow (New England Biolabs). Next, MBP was cleaved with the human rhinovirus 3C protease. The eluates were then desalted with 20 mM Tris-HCl, pH 8.0, and 150 mM NaCl utilizing a Bio-Scale Mini Bio-Gel P-6 desalting column (Bio-Rad Laboratories). Subsequently, the cleaved MBP was removed by reapplying FIP200 (1–634) to an Amylose Resin High Flow column. For the pull-down assay, MBP-FIP200 (1–634) eluted from the amylose resin continued to be purified on a HiLoad 26/60 Superdex 200 PG column eluted with 20 mM Tris-HCl, pH 8.0, and 150 mM NaCl.

Affinity chromatography with Amylose Resin High Flow (New England Biolabs) was used to purify the MBP-ATG13 (363–517) and MBP-ULK1 (636–1050) proteins. They were further purified on a HiLoad 26/60 Superdex 200 PG column and eluted with a buffer containing 20 mM Tris-HCl, pH 8.0, and 150 mM NaCl. Similarly, GST-ULK1 (636–1050) underwent initial purification using a GST-accept resin (Nacalai Tesque, 09277-14). This step was followed by further purification on a HiLoad 26/60 Superdex 200 PG column using the same elution buffer as above.

## In vitro pull-down assay

Purified proteins were incubated with GST-accept beads (Nacalai Tesque) at 4°C for 30 min. The beads were washed three times with PBS, and proteins were eluted with 10 mM glutathione in 50 mM Tris-HCl (pH 8.0). SDS–PAGE was used to separate the samples, and protein bands were detected by One Step CBB (BIO CRAFT).

## Isothermal titration calorimetry

The binding of FIP200 (1–634) to MBP-ATG13 (363–517) and MBP-ATG13 (363–517) to MBP-ULK1 (636–1050) was measured by ITC, with a MICROCAL PEAQ-ITC calorimeter (Malvern) and stirring at 750 rpm at 25°C. All proteins were prepared in a solution of 20 mM Tris-HCl, pH 8.0, and 150 mM NaCl. The titration of MBP-ATG13 (363–517) with FIP200 (1–634) involved 18 injections of 2 μl of the MBP-ATG13 (363–517) solution (229 μM) at 150-s intervals into a sample cell containing 300 μl of FIP200 (1–634) (4 μM). The titration of MBP-ULK1 (636–1050) with MBP-ATG13 (363–517) involved 18 injections of 2 μl of the MBP-ULK1 (636–1050) WT at 232 μM or FIP2A mutant at 172 μM at 150-s intervals into a sample cell containing 300 μl of MBP-ATG13 (363–517) at 22.9 or 20.0 μM, respectively. The isotherm was integrated and fitted with the one-side-binding model of the Malvern MicroCal PEAQ-ITC analysis software. The error of each parameter indicates the fitting error.

## Generation of KI cell lines

HeLa cells were transfected with the PX458-based plasmid expressing sgRNA for the *ATG13* gene and donor plasmids using FuGENE HD (E2311; Promega). One day after transfection, the KI cells were selected with 1 mg/ml G418 for 14 days. After single clones were isolated, clones positive for the FLAG tag and negative for wild-type ATG13 were screened by immunoblotting.

## Preparation of whole-cell lysates and immunoblotting

HeLa cells were lysed with 0.2% n-octyl-β-D-dodecyl maltoside in lysis buffer [50 mM Tris-HCl (pH 7.5), 150 mM NaCl, 1 mM EDTA, EDTA-free protease inhibitor cocktail (04080; Nacalai Tesque)] for 15 min on ice. After centrifugation at 20,000 × g for 15 min at 4°C, the supernatants were mixed with 20% volumes of 6 × SDS–PAGE sample buffer. The samples were subjected to SDS–PAGE and transferred to Immobilon-P PVDF membranes (IPVH00010; EMD Millipore). The primary antibodies used for immunoblotting were rabbit polyclonal antibodies against FIP200 (17250-1-AP; ProteinTech), ATG13 (13273; Cell Signaling), ULK1 (8054; Cell Signaling), ATG14 p-Ser29 (#92340; Cell Signaling), ATG14 (24412-1-AP; ProteinTech), and mouse monoclonal antibody against FLAG (F1804; Sigma-Aldrich) and β-actin (A2228; Sigma-Aldrich). Horseradish peroxidase (HRP)-conjugated anti-rabbit IgG (111-035-144; Jackson ImmunoResearch) was used as a secondary antibody. The HRP-conjugated mouse monoclonal antibody against DDDDK-tag (M185-7; MBL) was used for FLAG tag detection. Immobilon Western Chemiluminescent HRP Substrate (P90715; EMD Millipore) was used to visualize the signals detected by an image analyzer (ImageQuant LAS 4000; Cytiva). Fiji software (ImageJ; National Institutes of Health) was used to adjust contrast and brightness and quantify the signals (*Schindelin et al., 2012*).

## Immunoprecipitation

Cells were lysed with 1% 3-[(3-cholamidopropyl)-dimethylammonium]-1-propanesulfonate (CHAPS) in lysis buffer and centrifuged at 20,000 × g for 15 min. The supernatants were incubated with anti-FLAG M2 affinity gel (A2220; Sigma-Aldrich) for 1 hr at 4°C with gentle rotation. The beads were washed three times in washing buffer (20 mM Tris-HCl, pH 8.0, 150 mM NaCl, 1 mM EDTA, and 0.5% CHAPS), and the proteins were eluted with the SDS–PAGE sample buffer.

## Immunostaining and fluorescence microscopy

Cells were grown on segmented cover glass (SCC-002; Matsunami Glass IND.), washed with PBS, and fixed with 4% paraformaldehyde (163-20145; Fujifilm Wako Pure Chemical Corporation) in PBS for 10 min. The cells were permeabilized with 50 μg/ml digitonin in PBS for 5 min and blocked with 3% BSA in PBS for 30 min. Primary antibodies in PBS and 3% BSA were added, and samples were incubated for 1 hr at room temperature. Cells were washed five times with PBS, incubated with secondary antibodies in PBS with 3% BSA for 1 hr at room temperature, and washed five times with PBS. Rabbit

polyclonal antibodies against FIP200 (17250-1-AP; ProteinTech) and mouse monoclonal antibodies against FLAG (F1804; Sigma-Aldrich) HA (M180-3, MBL) and guinea pig antibody against p62 (GP62-C; PROGEN) were used as primary antibodies for immunostaining. Alexa Fluor 488-conjugated anti-mouse IgG (also cross-adsorbed to rabbit IgG and rat IgG) (A-11029; Thermo Fisher Scientific), Alexa Fluor 555-conjugated anti-rabbit IgG (also cross-adsorbed to mouse IgG) (A-31572; Thermo Fisher Scientific), and Alexa Fluor 647-conjugated anti-guinea pig (A21450; Thermo Fisher Scientific) were used as secondary antibodies. Cells were observed under a confocal microscope (ECLIPSE Ti2; Nikon).

## Halo-LC3 processing assay

Cells were treated with 100 nM tetramethylrhodamine (TMR)-conjugated HaloTag ligand (G8251, Promega) for 15 min. After washing twice with PBS, cells were cultured in the starvation medium for 1 hr and lysed as described above. Proteins were separated by SDS–PAGE, and in-gel TMR fluorescence was detected using a fluorescent imaging system (Odyssey M; Li-COR).

## Statistical analysis

GraphPad Prism 9 software (GraphPad Software) was used for statistical analyses. The statistical methods are described in each figure legend.

## Acknowledgements

We would like to thank Craig B Thompson for providing the *Ulk1/2* DKO MEFs; Shoji Yamaoka for the pMRXIP plasmid; Teruhito Yasui for the pCG-gag-pol and pCG-VSV-G plasmids; Sekiko Kurazono and Elijah William Caldwell for assistance with protein preparation. This work was supported in part by JSPS KAKENHI Grant Numbers JP19H05707, JP23K20044, JP23K06667, JP24H00060, JP25H00966, JP25H01320, and JP25H01321 (to NNN), JP23K14140, JP22KJ0044 (to YH), JP21H05731, JP23H02429, JP23K27122, JP23H04923 (to YF), JP21H05256 (to HY), JP22H04919 (to NM), CREST, Japan Science and Technology Agency Grant number JPMJCR20E3 (to NNN), ERATO, Japan Science and Technology Agency Grant number JPMJER1702 (to NM), PRIME, Japan Agency for Medical Research and Development Grant number JP20gm6410009 (to YF), and grants from the Takeda Science Foundation (to NNN, YF).

## Additional information

### Competing interests

Noboru Mizushima: Reviewing editor, eLife. The other author declares that no competing interests exist.

### Funding

| Funder | Grant reference number | Author |
| --- | --- | --- |
| Japan Society for the Promotion of Science | JP19H05707 | Nobuo N Noda |
| Japan Society for the Promotion of Science | JP23K20044 | Nobuo N Noda |
| Japan Society for the Promotion of Science | JP23K06667 | Nobuo N Noda |
| Japan Society for the Promotion of Science | JP24H00060 | Nobuo N Noda |
| Japan Society for the Promotion of Science | JP25H00966 | Nobuo N Noda |
| Japan Society for the Promotion of Science | JP25H01320 | Nobuo N Noda |
| Japan Society for the Promotion of Science | JP25H01321 | Nobuo N Noda |

| Funder | Grant reference number | Author |
|---|---|---|
| Japan Society for the Promotion of Science | JP23K14140 | Yutaro Hama |
| Japan Society for the Promotion of Science | JP22KJ0044 | Yutaro Hama |
| Japan Society for the Promotion of Science | JP21H05731 | Yuko Fujioka |
| Japan Society for the Promotion of Science | JP23H02429 | Yuko Fujioka |
| Japan Society for the Promotion of Science | JP23K27122 | Yuko Fujioka |
| Japan Society for the Promotion of Science | JP23H04923 | Yuko Fujioka |
| Japan Society for the Promotion of Science | JP21H05256 | Hayashi Yamamoto |
| Japan Society for the Promotion of Science | JP22H04919 | Noboru Mizushima |
| Japan Science and Technology Agency | 10.52926/jpmjcr20e3 | Nobuo N Noda |
| Japan Science and Technology Agency | 10.52926/jpmjer1702 | Noboru Mizushima |
| Japan Agency for Medical Research and Development | JP20gm6410009 | Yuko Fujioka |
| Takeda Science Foundation | | Yuko Fujioka Nobuo Noda |

The funders had no role in study design, data collection, and interpretation, or the decision to submit the work for publication.

## Author contributions

Yutaro Hama, Conceptualization, Data curation, Formal analysis, Funding acquisition, Validation, Investigation, Visualization, Methodology, Writing – original draft; Yuko Fujioka, Data curation, Formal analysis, Funding acquisition, Validation, Investigation; Hayashi Yamamoto, Noboru Mizushima, Funding acquisition, Validation, Investigation; Nobuo N Noda, Conceptualization, Data curation, Formal analysis, Supervision, Funding acquisition, Validation, Visualization, Writing – original draft, Writing – review and editing

## Author ORCIDs

Yutaro Hama ⓘ https://orcid.org/0000-0002-7190-4547
Yuko Fujioka ⓘ https://orcid.org/0000-0002-6905-0669
Hayashi Yamamoto ⓘ https://orcid.org/0000-0002-2831-1463
Noboru Mizushima ⓘ https://orcid.org/0000-0002-6258-6444
Nobuo N Noda ⓘ https://orcid.org/0000-0002-6940-8069

Reviewer #1 (Public review): https://doi.org/10.7554/eLife.101531.3.sa1
Reviewer #2 (Public review): https://doi.org/10.7554/eLife.101531.3.sa2
Reviewer #3 (Public review): https://doi.org/10.7554/eLife.101531.3.sa3
Author response https://doi.org/10.7554/eLife.101531.3.sa4

# Additional files

## Supplementary files

MDAR checklist

## Data availability

All the data underlying this study are available in the published article and its supplementary material.

The following dataset was generated:

| Author(s) | Year | Dataset title | Dataset URL | Database and Identifier |
|---|---|---|---|---|
| Noda NN | 2025 | ULK1 complex | https://modelarchive.org/doi/10.5452/ma-jz53c | ModelArchive, ma-jz53c |

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
