## [Editor Report · eLife Assessment]

Building on previous structural studies, this work provides **valuable** new insights into the architecture of the autophagy initiation complex, comprising ULK1, ATG13, and FIP200. The authors present their findings with **solid** supporting evidence, making this study a significant contribution to the autophagy field.

---

## [Referee Report · Reviewer #1 (Public review)]

In this study, Hama et al. investigated the molecular regulatory mechanisms underlying the formation of the ULK1 complex in mammalian cells. Their results showed that in mammalian cells, ULK1, ATG13, and FIP200 form a complex with a stoichiometry of 1:1:2. These predicted interaction regions were validated through both in vivo and in vitro experiments, providing deeper insight into the molecular basis of ULK1 complex assembly in mammalian cells.

The revised manuscript has addressed the majority of my concerns, and I have no further questions. Overall, this is a solid and impactful study that significantly advances our understanding of how the ULK1 complex is formed.

---

## [Referee Report · Reviewer #2 (Public review)]

Summary:

This is important work that helps to uncover how the process of autophagy is initiated - via structural analyses of the initiating ULK1 complex. High resolution structural details and a mechanistic insight of this complex have been lacking and understanding how it assembles and functions is a major goal of a field that impacts many aspects of cell and disease biology. While we know components of the ULK1 complex are essential for autophagy, how they physically interact is far from clear. The work presented makes use of AlphaFold2 to structurally predict interaction sites between the different subunits of the ULK1 complex (namely ULK1, ATG13 and FIP200). Importantly, the authors go on to experimentally validate that these predicted sites are critical for complex formation by using site-directed mutagenesis and then go on to show that the three-way interaction between these components is necessary to induce autophagy in cells.

Strengths:

The data are very clear. Each binding interface of ATG13 (ATG13 with FIP300/ATG13 with ULK1) is confirmed biochemically with ITC and IP experiments from cells. Likewise, IP experiments with ULK1 and FIP200 also validate interaction domains. A real strength of the work is in the analyses of the consequences of disrupting ATG13's interactions in cells. The authors make CRISPR KI mutations of the binding interface point mutants. This is not a trivial task and is the best approach as everything is monitored under endogenous conditions. Using these cells the authors show that ATG13's ability to interact with both ULK1 and FIP200 is essential for a full autophagy response.

Weaknesses:

I think a main weakness here is the failure to acknowledge and compare results with an earlier preprint that shows essentially the same thing (https://doi.org/10.1101/2023.06.01.543278). Arguably, this earlier work is much stronger from a structural point of view as it relies not only on AlphaFold2 but also actual experimental structural determinations (and takes the mechanisms of autophagy activation further by providing evidence for a super complex between the ULK1 and VPS34 complexes). That is not to say that this work is not important, as in the least it independently helps to build a consensus for ULK1 complex structure. Another weakness is that the downstream "functional" consequences of disrupting the ULK1 complex are only minimally addressed. The authors perform a Halotag-LC3 autophagy assay, which essentially monitors the endpoint of the process. There are a lot of steps in between, knowledge of which could help with mechanistic understanding. Not in the least is the kinase activity of ULK1 - how is this altered by disrupting its interactions with ATG13 and/or FIP200?

Update:

I feel the authors have addressed my concerns in their revised manuscript

---

## [Referee Report · Reviewer #3 (Public review)]

In this study, the authors employed the protein complex structure prediction tool AlphaFold-Multimer to obtain a predicted structure of the protein complex composed of ULK1-ATG13-FIP200 and validated the structure using mutational analysis. This complex plays a central role in the initiation of autophagy in mammals. The results obtained in this study reveal extensive binary interactions between ULK1 and ATG13, between ULK1 and FIP200, and between ATG13 and FIP200, and pinpoint the critical residues at each interaction interface. Mutating these critical residues led to the loss of binary interactions. Interestingly, the authors showed that the ATG13-ULK1 interaction and the ATG13-FIP200 interaction are partially redundant for maintaining the complex. The experimental data presented by the authors are of high quality and convincing. The revised manuscript offers enhanced details about the prediction procedure and results, along with additional experimental findings, significantly increasing the scientific value of this paper.

---

## [Author Response]

The following is the authors’ response to the original reviews

**Public Reviews:**

**Reviewer #1 (Public review):**
In this study, Hama et al. explored the molecular regulatory mechanisms underlying the formation of the ULK1 complex. By employing the AlphaFold structural prediction tool, they showed notable differences in the complex formation mechanisms between ULK1 in mammalian cells and Atg1 in yeast cells. Their findings revealed that in mammalian cells, ULK1, ATG13, and FIP200 form a complex with a stoichiometry of 1:1:2. These predicted interaction regions were validated through both in vivo and in vitro assays, enhancing our understanding of the molecular mechanisms governing ULK1 complex formation in mammalian cells. Importantly, they identified a direct interaction between ULK1 and FIP200, which is crucial for autophagy. However, some aspects of this manuscript require further clarification, validation, and correction by the authors.

Thank you for your thorough evaluation of our manuscript. We have carefully revised the manuscript to address your concerns by performing extra experiments and providing additional clarifications, validations, and corrections as written below.

**Reviewer #2 (Public review):**
Summary:This is important work that helps to uncover how the process of autophagy is initiated - via structural analyses of the initiating ULK1 complex. High-resolution structural details and a mechanistic insight of this complex have been lacking and understanding how it assembles and functions is a major goal of a field that impacts many aspects of cell and disease biology. While we know components of the ULK1 complex are essential for autophagy, how they physically interact is far from clear. The work presented makes use of AlphaFold2 to structurally predict interaction sites between the different subunits of the ULK1 complex (namely ULK1, ATG13, and FIP200). Importantly, the authors go on to experimentally validate that these predicted sites are critical for complex formation by using site-directed mutagenesis and then go on to show that the three-way interaction between these components is necessary to induce autophagy in cells.Strengths:The data are very clear. Each binding interface of ATG13 (ATG13 with FIP300/ATG13 with ULK1) is confirmed biochemically with ITC and IP experiments from cells. Likewise, IP experiments with ULK1 and FIP200 also validate interaction domains. A real strength of the work in in their analyses of the consequences of disrupting ATG13's interactions in cells. The authors make CRISPR KI mutations of the binding interface point mutants. This is not a trivial task and is the best approach as everything is monitored under endogenous conditions. Using these cells the authors show that ATG13's ability to interact with both ULK1 and FIP200 is essential for a full autophagy response.

Thank you for your thoughtful review and for highlighting the importance of our approach.

Weaknesses:I think a main weakness here is the failure to acknowledge and compare results with an earlier preprint that shows essentially the same thing (https://doi.org/10.1101/2023.06.01.543278). Arguably this earlier work is much stronger from a structural point of view as it relies not only on AlphaFold2 but also actual experimental structural determinations (and takes the mechanisms of autophagy activation further by providing evidence for a super complex between the ULK1 and VPS34 complexes). That is not to say that this work is not important, as in the least it independently helps to build a consensus for ULK1 complex structure. Another weakness is that the downstream "functional" consequences of disrupting the ULK1 complex are only minimally addressed. The authors perform a Halotag-LC3 autophagy assay, which essentially monitors the endpoint of the process. There are a lot of steps in between, knowledge of which could help with mechanistic understanding. Not in the least is the kinase activity of ULK1 - how is this altered by disrupting its interactions with ATG13 and/or FIP200?

Thank you for this valuable feedback. In response, we performed a detailed structural comparison between the cryo-EM structure reported in the referenced preprint and our AlphaFold-based model. We have summarized both the similarities and differences in newly included figures (revised Figure 2A, B, 3B, S1F) and provided an in-depth discussion in the main text. Furthermore, to address the downstream consequences of ULK1 complex disruption, we have investigated the impact on ULK1 kinase activity, specifically examining how mutations affecting ATG13 or FIP200 interaction alter ULK1’s phosphorylation of a key substrate ATG14. In addition, we analyzed the effect on ATG9 vesicle recruitment. We provide the corresponding data as Figure S3C-E and detailed discussions in the revised manuscript.

**Reviewer #3 (Public review):**
In this study, the authors employed the protein complex structure prediction tool AlphaFold-Multimer to obtain a predicted structure of the protein complex composed of ULK1-ATG13-FIP200 and validated the structure using mutational analysis. This complex plays a central role in the initiation of autophagy in mammals. Previous attempts at resolving its structure have failed to obtain high-resolution structures that can reveal atomic details of the interactions within the complex. The results obtained in this study reveal extensive binary interactions between ULK1 and ATG13, between ULK1 and FIP200, and between ATG13 and FIP200, and pinpoint the critical residues at each interaction interface. Mutating these critical residues led to the loss of binary interactions. Interestingly, the authors showed that the ATG13-ULK1 interaction and the ATG13-FIP200 interaction are partially redundant for maintaining the complex.

We are grateful for your high evaluation of our work.

The experimental data presented by the authors are of high quality and convincing. However, given the core importance of the AlphaFold-Multimer prediction for this study, I recommend the authors improve the presentation and documentation related to the prediction, including the following:(1) I suggest the authors consider depositing the predicted structure to a database (e.g. ModelArchive) so that it can be accessed by the readers.

We have deposited the AlphaFold model to ModelArchive with the accession code ma-jz53c, which is indicated in the revised manuscript.

(2) I suggest the authors provide more details on the prediction, including explaining why they chose to use the 1:1:2 stoichiometry for ULK1-ATG13-FIP200 and whether they have tried other stoichiometries, and explaining why they chose to use the specific fragments of the three proteins and whether they have used other fragments.

We appreciate your suggestion. As we noted in the original manuscript, previous studies have shown that the C-terminal region of ULK1 and the C-terminal intrinsically disordered region of ATG13 bind to the N-terminal region of the FIP200 homodimer (Alers, Loffler et al., 2011; Ganley, Lam du et al., 2009; Hieke, Loffler et al., 2015; Hosokawa, Hara et al., 2009; Jung, Jun et al., 2009; Papinski and Kraft, 2016; Wallot-Hieke, Verma et al., 2018). We relied on these findings when determining the specific regions to include in our complex prediction and when selecting a 1:1:2 stoichiometry for ULK1–ATG13–FIP200 which was reported previously (Shi et al., 2020). We also used AlphaFold2 to predict the structures of the full-length ULK1–ATG13 complex and the complex of the FIP200N dimer with full-length ATG13, confirming that there were no issues with our choice of regions (revised Figure S1A-C). In the revised manuscript, we have provided a more detailed explanation of our rationale based on the previous reports and additional AlphaFold predictions.

(3) I suggest the authors present the PAE plot generated by AlphaFold-Multimer in Figure S1. The PAE plot provides valuable information on the prediction.

We provided the PAE plot in the revised Figure S1C.

**Recommendations for the authors:**

**Reviewer #1 (Recommendations for the authors):**
(1) In Figure 1D, the labels for the input and IP of ATG13-FLAG should be corrected to ATG13-FLAG FIP3A.

We thank the reviewer for pointing out these labeling mistakes. We revised the labels based on the suggestions.

(2) In the discussion section, the authors should address why ATG13-FLAG ULK1 2A in Fig. 2D leads to a significantly lower expression of ULK1 and provide possible explanations for this observation.

ATG13 and ATG101, both core components of the ULK1 complex, are known to stabilize each other through their mutual interaction. Loss or reduction of one protein typically leads to the destabilization of the other. In this context, ULK1 is similarly stabilized by binding to ATG13. Therefore, ATG13-FLAG ULK2A mutant, which has reduced binding to ULK1, likely loses this stabilizing activity and ULK1 becomes destabilized, resulting in the lower expression levels of ULK1. We added these discussions in the revised manuscript.

(3) In Figure 4B, the authors should explain why Atg13-FLAG KI significantly affects the expression of endogenous ULK1. Could Atg13-FLAG KI be interfering with its binding to ULK1? Experimental evidence should be provided to support this. Additionally, does Atg13-FLAG KI affect autophagy? Wild-type HeLa cells should be included as a control in Figure 4C and 4D to address this question.

Thank you for your constructive suggestion. We found a technical error in the ULK1 blot of Figure 4B. Therefore, we repeated the experiment. The results show that ULK1 expression did not significantly change in the ATG13-FLAG KI. These findings are consistent with Figure S3A. We have replaced Figure 4B with this new data.

We agree that including wild-type HeLa cells as a control is essential to determine whether ATG13-FLAG KI affects autophagy. We performed the same experiments in wild-type HeLa cells and found that ATG13-FLAG KI does not significantly impact autophagic flux. Accordingly, we have replaced Figures 4D and 4E with these new data.

(4) In Figure 3C, the authors used an in vitro GST pulldown assay to detect a direct interaction between ULK1 and FIP200, which was also confirmed in Figure 3E. However, since FLAG-ULK1 FIP2A affects its binding with ATG13 (Fig. 3E), it is possible that ULK1 FIP2A inhibits autophagy by disrupting this interaction. The authors should therefore use an in vitro GST pulldown assay to determine whether GST-ULK1 FIP2A affects its binding with ATG13. Additionally, the authors should investigate whether the interaction between ULK1 and FIP200 in cells requires the involvement of ATG13 by using ATG13 knockout cells to confirm if the ULK1-FIP200 interaction is affected in the absence of ATG13.

Thank you for the valuable suggestion. We examined the effect of the FIP2A mutation on the ULK1–ATG13 interaction using isothermal titration calorimetry (ITC) to obtain quantitative binding data. The results showed that the FIP2A mutation does not markedly alter the affinity between ULK1 and ATG13 (revised Figure S2B), suggesting that FIP2A mainly weakens the ULK1–FIP200 interaction. Regarding experiments in ATG13 knockout cells, ULK1 becomes destabilized in the absence of ATG13, making it technically difficult to assess how the ULK1–FIP200 interaction is affected under those conditions.

**Reviewer #2 (Recommendations for the authors):**
I feel the manuscript would benefit from a more detailed comparison with the Hurely lab paper - are the structural binding interfaces the same, or are there differences?

We appreciate the suggestion to compare our results more closely with the work from the Hurley lab. We performed a detailed structural comparison between the cryo-EM structure reported in the referenced preprint and our AlphaFold-based model (revised Figure 2A, B, 3B, S1F) and provided an in-depth discussion in the main text.

As mentioned, what happens downstream of disrupting the ULK1 complex? How is ULK1 activity changed, both in vitro and in cells? Does disruption of the ULK1 complex binding sites impair VPS34 activity in cells (for example by looking at PtdIns3P levels/staining)?

Thank you for your insightful comments. We focused on elucidating how disrupting the ULK1 complex leads to impaired autophagy. To assess ULK1 activity, we measured ULK1-dependent phosphorylation of ATG14 at Ser29 (PMID: 27046250; PMID: 27938392). In FIP3A and FU5A knock-in cells, ATG14 phosphorylation was significantly reduced, indicating decreased ULK1 activity (revised Figure S3D, E). This observation is consistent with previous work showing that FIP200 recruits the PI3K complex. Notably, in ATG13 knockout cells, ATG14 phosphorylation became almost undetectable, though the underlying mechanism remains to be fully investigated. Altogether, these data point to reduced ULK1 activity as a key factor explaining the autophagy deficiency observed in FU5A knock-in cells.

We also explored possible downstream mechanisms. One well-established function of ATG13 is to recruit ATG9 vesicles (PMID: 36791199). These vesicles serve as an upstream platform for the PI3K complex, providing the substrate for phosphoinositide generation (PMID: 38342428). To clarify how our mutations impact this step, we starved ATG13-FLAG knock-in cells and observed ATG9 localization. Unexpectedly, even in FU5A knock-in cells where ATG13 is almost completely dissociated from the ULK1 complex, ATG9A still colocalized with FIP200 (revised Figure S3C). These puncta also overlapped with p62, likely because p62 bodies recruit both FIP200 and ATG9 vesicles. Although we suspect that ATG9 recruitment is nonetheless impaired under these conditions, we were unable to definitively demonstrate this experimentally and consider it an important avenue for future study.

**Reviewer #3 (Recommendations for the authors):**
Here are some additional minor suggestions:(1) The UBL domains are only mentioned in the abstract but not anywhere else in the manuscript. I suggest the authors add descriptions related to the UBL domains in the Results section.

We thank the reviewer for pointing out the lack of description of UBL domains, which we added in Results in the revised manuscript.

(2) The authors may want to consider adding a diagram in Figure 1A to show the domain organization of the three full-length proteins and the ranges of the three fragments in the predicted structure.

We have added a proposed diagram as Figure 1A.

(3) I suggest the authors consider highlighting in Figure 1A the positions of the binding sites shown in Figure 1B, for example, by adding arrows in Figure 1A.

We have added arrows in the revised Figure 1B (which was Figure 1A in the original submission).

(4) In Figure 1D, "Atg13-FLAG" should be "Atg13-FLAG FIP3A".

We have revised the labeling in Figure 1D.

(5) "the binding of ATG13 and ULK1 to the FIP200 dimer one by one" may need to be re-phrased. "One by one" conveys a meaning of "sequential", which is probably not what the authors meant to say.

We have revised the sentence as “the binding of one molecule each of ATG13 and ULK1 to the FIP200 dimer”.

(6) In "Wide interactions were predicted between the four molecules", I suggest changing "wide" to "extensive".

We have changed “wide” to “extensive” in the revised manuscript.

(7) In "which revealed that the tandem two microtubule-interacting and transport (MIT) domains in Atg1 bind to the tandem two MIT interacting motifs (MIMs) of ATG13", I suggest changing the two occurrences of "tandem two" to "two tandem" or simply "tandem".

We simply used "tandem" in the revised manuscript.